# EGFR-MEK1/2 cascade negatively regulates bactericidal function of bone marrow macrophages in mice with *Staphylococcus aureus* osteomyelitis

**Mingchao Jin**[1,2☯], **Xiaohu Wu**[1,2☯], **Jin Hu**[1,2☯], **Yijie Chen**[1,2], **Bingsheng Yang**[1,2], **Chubin Cheng**[1,2], **Mankai Yang**[1,2], **Xianrong Zhang**[1,2]*

**1** Division of Orthopaedics and Traumatology, Department of Orthopaedics, Nanfang Hospital, Southern Medical University, No.1838 North of Guangzhou Avenue, Guangzhou, Guangdong Province, China, **2** Guangdong Provincial Key Laboratory of Bone and Cartilage Regenerative Medicine, Nanfang Hospital, Southern Medical University, Guangzhou, China

☯ These authors contributed equally to this work.
* xianrongzh@smu.edu.cn

**Data Availability Statement:** Transcriptome data are available in Gene Expression Omnibus (GEO) under the accession number GSE272198 (https://

## Abstract

The ability of *Staphylococcus aureus* (*S. aureus*) to survive within macrophages is a critical strategy for immune evasion, contributing to the pathogenesis and progression of osteomyelitis. However, the underlying mechanisms remain poorly characterized. This study discovered that inhibiting the MEK1/2 pathway reduced bacterial load and mitigated bone destruction in a mouse model of *S. aureus* osteomyelitis. Histological staining revealed increased phosphorylated MEK1/2 levels in bone marrow macrophages surrounding abscess in the mouse model of *S. aureus* osteomyelitis. Activation of MEK1/2 pathway and its roles in impairing macrophage bactericidal function were confirmed in primary mouse bone marrow-derived macrophages (BMDMs). Transcriptome analysis and *in vitro* experiments demonstrated that *S. aureus* activates the MEK1/2 pathway through EGFR signaling. Moreover, we found that excessive activation of EGFR-MEK1/2 cascade downregulates mitochondrial reactive oxygen species (mtROS) levels by suppressing Chek2 expression, thereby impairing macrophage bactericidal function. Furthermore, pharmacological inhibition of EGFR signaling prevented upregulation of phosphorylated MEK1/2 and restored Chek2 expression in macrophages, significantly enhancing *S. aureus* clearance and improving bone microstructure *in vivo*. These findings highlight the critical role of the EGFR-MEK1/2 cascade in host immune defense against *S. aureus*, suggesting that *S. aureus* may reduce mtROS levels by overactivating the EGFR-MEK1/2 cascade, thereby suppressing macrophage bactericidal function. Therefore, combining EGFR-MEK1/2 pathway blockade with antibiotics could represent an effective therapeutic approach for the treatment of *S. aureus* osteomyelitis.

www.ncbi.nlm.nih.gov/geo/query/acc.cgi?acc=
GSE272198). All the other data are available in the
article and supporting information.

**Funding:** This work was supported by the National
Natural Science Foundation of China (No.
82072459 to X.Z. and No. 82272258 to X.Z.). The
funders had no role in study design, data collection
and analysis, decision to publish, or preparation of
the manuscript.

**Competing interests:** The authors have declared
that no competing interests exist.

## Author summary

Osteomyelitis, characterized by progressive inflammation and tissue destruction in bone,
is predominantly caused by *Staphylococcus aureus* (*S. aureus*). However, treating *S. aureus*
osteomyelitis remains a major clinical challenge due to its ability to survive in phagocytes
like macrophages, evading host immune surveillance and antibiotic treatments, leading to
recurrent and persistent infections. This study aimed to elucidate a potential molecular
mechanism underlying the *S. aureus* survival in macrophages. Our findings demonstrate
that *S. aureus* induces phosphorylation of MEK1/2 in macrophages localized at infection
sites. Inhibiting either the MEK1/2 pathway or EGFR signaling enhances bacterial clear-
ance both *in vivo* and *in vitro*. Persistent *S. aureus* infection was found to suppress Chek2
expression through the activation of the EGFR-MEK1/2 pathway, leading to decreased
production of mitochondrial ROS and impaired macrophage bactericidal activity. This
study highlights the critical role of EGFR-MEK1/2 pathway in macrophage antimicrobial
dysfunction and the pathogenesis of *S. aureus* osteomyelitis. Targeting this pathway in
conjunction with antibiotics may offer a promising therapeutic strategy in *S. aureus*
osteomyelitis.

## Introduction

Osteomyelitis, characterized by progressive inflammation and destruction in bone and sur-
rounding tissue caused by microbial pathogens, is a challenging condition in orthopedics [1].
*Staphylococcus aureus* (*S. aureus*) is the most prevalent causative pathogen isolated in cases of
osteomyelitis. Due to its resilience, *S. aureus* is closely associated with recurrent osteomyelitis
[2–4]. Even after surgical debridement, implant removal, and a full course of antibiotics, treat-
ment failure rates can reach 37.9%, with a recurrence rate of 25.9% [3,5]. Therefore, new thera-
peutic strategies are needed to overcome *S. aureus* therapy resistance in osteomyelitis.

The ability of *S. aureus* to evade host immune surveillance is critical for its persistence and
relapse despite antibiotic treatment. This pathogen employs various strategies, such as produc-
ing toxins to hijack host cell signaling, forming biofilms on implants and in the osteocyte
lacuno-canalicular network of cortical bone, and internalizing into both phagocytic and non-
phagocytic cells [6–10]. Macrophages and neutrophils are recruited to infection sites to eradi-
cate bacteria, forming the first line of host defense against *S. aureus* [11,12]. However,
accumulating evidence suggests that *S. aureus* can survive intracellularly within macrophages,
contributing to bacterial dissemination and pathogenesis [13,14]. Understanding the mecha-
nisms that enable *S. aureus* survival within macrophages is essential.

The extracellular signal-regulated kinase (ERK)/mitogen-activated protein kinases
(MAPKs), a classical MAPKs cascade comprising MAPKK kinase, MAPK kinase 1/2 (MEK1/
2), and ERK1/2, is well known for its role in cellular proliferation, differentiation, and survival
[15]. Accumulating evidence suggests that aberrant activation of the MEK1/2-ERK1/2 pathway
supports the pathogenic process during viral infection and is closely associated with the pro-
gression of inflammatory tissue injury [16–18]. A recent study showed that combining antibi-
otic treatment with MEK1/2-p-ERK1/2 inhibition reduced excessive inflammation and
restored bone union in a murine MRSA-infected fracture model [19]. However, the role of the
MEK1/2 pathway in the immune defense against *S. aureus* remains unclear.

Here, we investigated the role of MEK1/2 pathway in the pathogenesis of *S. aureus* osteo-
myelitis and explored how its overactivation might suppress macrophage bactericidal activity.
We found phosphorylated MEK1/2 (p-MEK1/2) levels were upregulated in bone marrow

macrophages during *S. aureus* infection both *in vivo* and *in vitro*. Blocking this pathway rescued bone destruction and reduced the bacterial burden in infected bones. In addition, we identified a novel function of the overactivated epidermal growth factor receptor (EGFR)-MEK1/2 cascade in suppressing mitochondrial ROS (mtROS) levels, thereby impairing macrophage bacterial clearance capacity. Our findings suggest that targeting the MEK1/2 pathway could be a promising strategy for overcoming resistance to therapy in *S. aureus*-induced osteomyelitis.

## Results

### Pharmacological blockade of MEK1/2 pathway alleviates bone destruction and bacterial load in femurs of mice with *S. aureus* osteomyelitis

Recent studies have documented that activation of the MEK1/2 pathway represents a crucial signal transducer in response to viral or bacterial infection, leading to inflammatory responses and tissue injury [18,19]. To investigate whether pharmacological blocking of the MEK1/2 pathway can inhibit the pathogenesis of *S. aureus* osteomyelitis, we treated mice with *S. aureus* osteomyelitis using PD0325901, an MEK1/2 inhibitor, in combination with gentamicin. As revealed by micro-CT imaging, PD0325901-treated mice exhibited significantly reduced cortical bone loss compared with vehicle-treated mice (Fig 1A and 1B). In addition, PD0325901-treated mice showed significant improvements in bone mineral density (BMD) and bone volume/tissue volume (BV/TV) in the distal femora compared with vehicle-treated mice (Fig 1C and 1D). Further analysis showed that the improved BV/TV in PD0325901-treated mice was mainly due to rescued trabecular number (Tb.N) and a significantly decreased trabecular bone pattern factor (Tb.Pf), with no obvious changes in the trabecular thickness (Tb.Th) (Fig 1E–1G). Consistent with these microstructural changes, H&E staining showed severe loss of trabecular bone density, empty lacunae in remaining trabeculae in distal femoral metaphysis, fibroblastic hyperplasia in peri-implant bone marrow, and extensive reactive new bone formation around cortical bone in femurs of vehicle-treated mice. In contrast, PD0325901-treated mice exhibited significantly improved bone morphology and structure with limited pathological changes around the implant (Fig 1H). Blind scoring of bone sections confirmed notably rescued bone destruction in *S. aureus*-infected femurs in PD0325901-treated mice compared with vehicle-treated mice (Fig 1I).

The observation that PD0325901 treatment alleviated bone destruction in mice with *S. aureus* osteomyelitis suggested that blocking the MEK1/2 pathway may reduce bacterial burden in bone. In support of this hypothesis, immunofluorescence assays showed significantly lower amounts of *S. aureus*-positive staining in bone marrow of PD0325901-treated mice compared with vehicle-treated mice (Fig 1J and 1K). In addition, classical culture-based analysis revealed a significant decrease in bacterial CFU per gram of bone tissue in infected femurs in PD0325901-treated mice compared with vehicle-treated mice (Fig 1L and 1M). Taken together, these results suggest that overactivation of the MEK1/2 pathway may impair antibacterial defense.

### Phosphorylation of MEK1/2 in macrophages is upregulated in mouse femurs following *S. aureus* osteomyelitis

MEK1/2 is typically activated via serine phosphorylation by upstream activator kinases [20]. To assess the effect of *S. aureus* infection on MEK1/2 activity, we analyzed the levels of phosphorylated MEK1/2 (p-MEK1/2) in mouse femurs 14 days post-infection. Immunostaining results revealed a notably elevated level of p-MEK1/2, mainly located in cells surrounding the

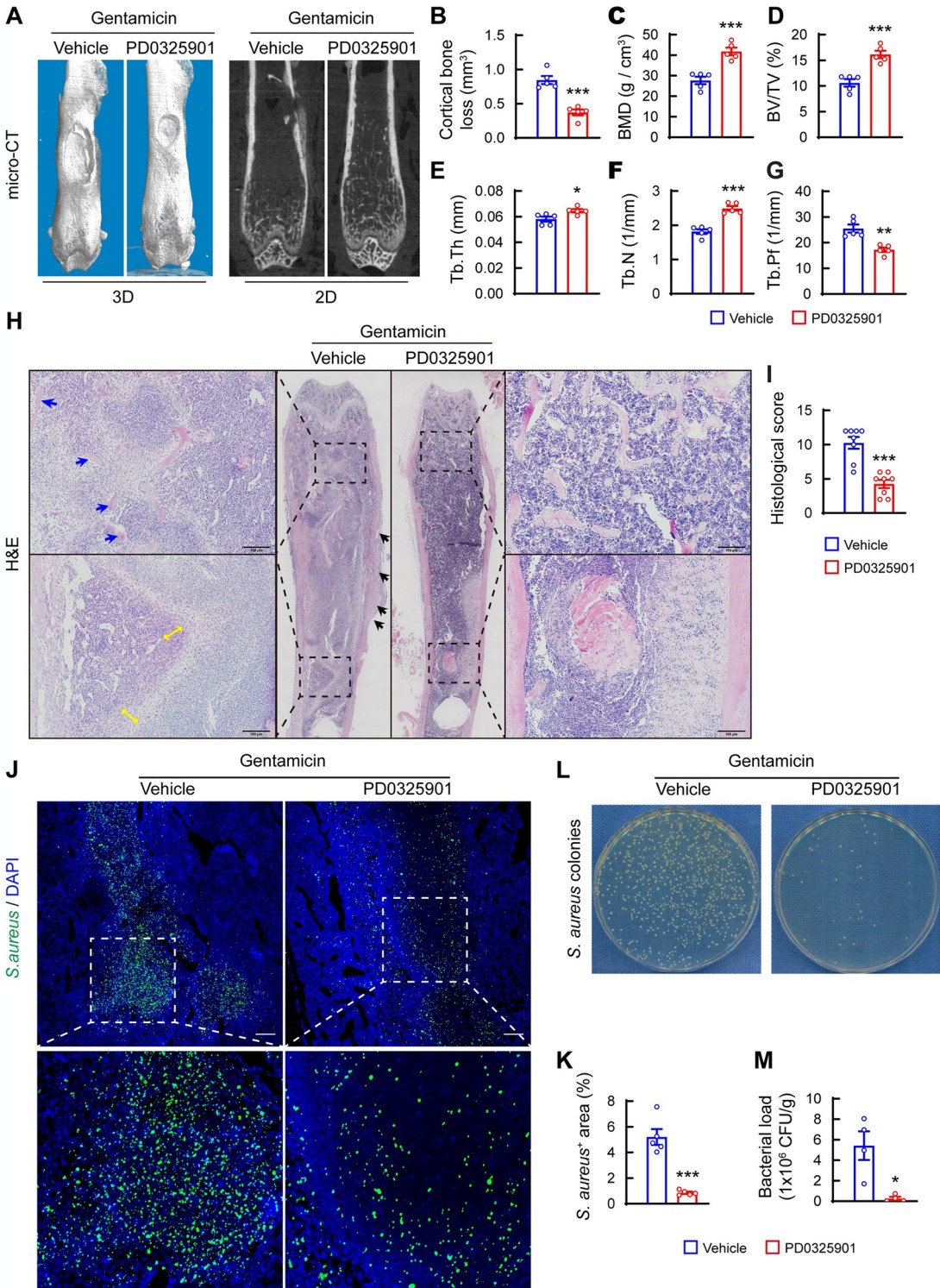

**Fig 1. Inhibition of MEK1/2 alleviates bone destruction and reduces bacterial burden in bone of mice with *S. aureus* osteomyelitis.** (A) Representative 3D and 2D micro-CT images and quantitative analysis of cortical bone loss (B), trabecular bone mineral density (BMD) (C), bone fraction (BV/TV) (D), trabecular thickness (Tb.Th) (E), trabecular number (Tb.N) (F), and trabecular bone pattern factor (Tb.Pf) (G). n = 5/group. (H) Representative images of H&E staining. Blue arrows indicate trabecular bone with empty lacunae, double orange arrows show fibroblastic hyperplasia in peri-implant bone marrow, and black arrows show reactive new bone formation around cortical bone. Scale bars, 100 μm. (I) Quantitative analysis of histopathological

changes using the scoring system established by Smeltzer et al [51]. n = 8/group, Mann-Whitney test. (J) Representative images and (K) Quantification of immunofluorescence for *S. aureus*-positive staining area in femurs in *S. aureus* osteomyelitis mice treated with PD0325901 (0.2 mg/kg/day) or the same volume of vehicle (5% DMSO). Scale bars, 200 μm. n = 5/group. (L) Representative images and (M) Quantification of *S. aureus* colonies growing on an agar plate after being recovered from the femur. 10 μl of *S. aureus* suspension diluted in PBS (1: 500) was placed and cultured on LB agar plates for 24 h at 37˚C. n = 5/group. *$p < 0.05$, **$p < 0.01$, ***$p < 0.001$, statistical significance was analyzed by Student's *t* test.

abscess in bone marrow (Fig 2A and 2B). Since neutrophils are the main immune cells in infectious nidus, surrounded by macrophages [21], we examined p-MEK1/2 expression in F4/80+ cells using immunofluorescence staining. Our data clearly showed a significant increase in F4/80+p-MEK1/2+ cells in bone marrow away from abscess in *S. aureus*-infected right femurs compared with control mice (Fig 2C and 2D). Notably, p-MEK1/2 levels were predominantly upregulated in F4/80+ macrophages surrounding the abscess (Fig 2C and 2D). Moreover, the levels of F4/80+p-MEK1/2+ cells in the left femoral bone marrow (the contralateral side of surgery and infection) of *S. aureus*-infected mice were comparable to those in control mice (S1 Fig). These data suggest that the MEK1/2 pathway is primarily activated in F4/80+ macrophages surrounding the abscess.

## Activation of the MEK1/2 pathway is associated with reduced mitochondrial ROS (mtROS) levels and suppressed bactericidal activity in macrophages

We next investigated whether *S. aureus* might activate the MEK1/2 pathway in primary cultures of BMDMs. Western blot analysis revealed that *S. aureus* upregulated p-MEK1/2 levels in BMDMs in a concentration-dependent manner after 30 min (Fig 3A and 3B). This observation suggests that the MEK1/2 pathway in macrophages may be rapidly activated in response to *S. aureus* infection.

We then assessed whether overactivation of the MEK1/2 pathway affects macrophage function during *S. aureus* infection. Blocking the MEK1/2 pathway with PD0325901 did not affect the phagocytic ability of BMDMs (Figs 3C and S2A). Importantly, the bacterial killing assay demonstrated that PD0325901 significantly reduced intracellular bacterial burden in BMDMs at indicated time points post-infection (Figs 3D and S2B), indicating enhanced bactericidal function of macrophages with MEK1/2 pathway inhibition.

Given the critical role of mtROS in the bactericidal function of macrophages [22], and our recent finding of a notable reduction in mtROS levels 12 h and 24 h post-infection [23], we determined whether inhibiting MEK1/2 pathway affects mtROS levels in macrophages. Mito-Tracker Green and MitoSOX staining revealed that PD0325901 treatment for 12 h or 24 h significantly increased mtROS levels in BMDMs (Fig 3E and 3F). These findings suggest that overactivation of the MEK1/2 pathway suppresses mtROS production, thereby impairing the bactericidal function of macrophages.

Considering the crucial role of phagocyte NADPH oxidase in host defense by generating superoxide anion and other ROS molecules [24,25], we evaluated the effect of PD0325901 on total protein levels of NADPH oxidase components, including p22phox, NOX2/gp91phox, p47phox, p67phox, and p40phox. Surprisingly, the levels of these proteins remained unchanged after 12 and 24 h of *S. aureus* infection (S2C–S2H Fig). However, PD0325901 treatment significantly upregulated NOX2 levels after 12 h and 24 h of *S. aureus* infection (Fig 3G and 3H). We inferred that the ability of BMDMs to produce ROS from NADPH oxidase in phagocyte decreased to basal levels, insufficient to restrict invasion of *S. aureus* after 12 h of persistent infection. Thus, overactivation of MEK1/2 may not only suppress mtROS

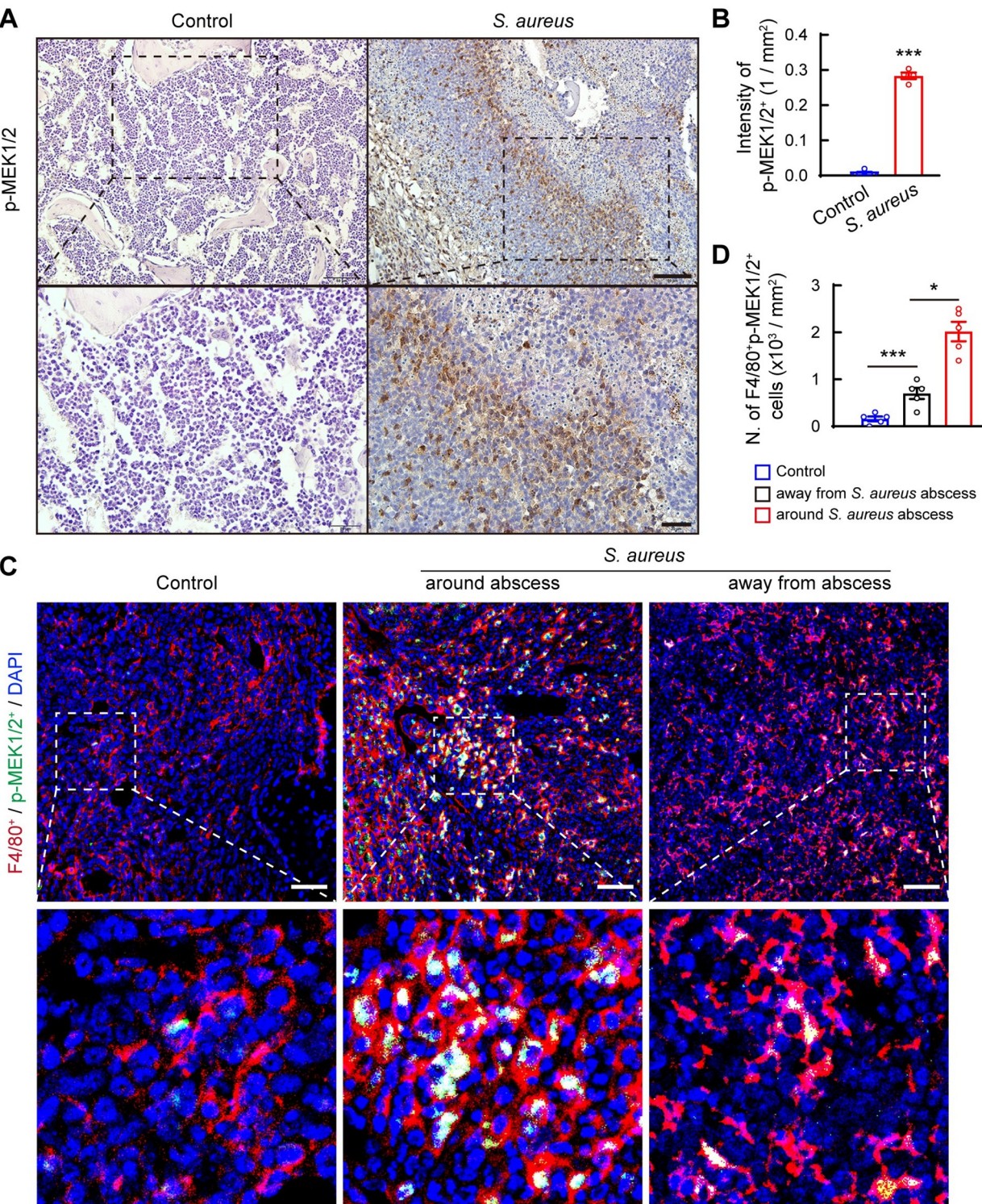

**Fig 2. The phosphorylation level of MEK1/2 is upregulated in macrophages surrounding bone marrow abscesses in mice with *S. aureus* osteomyelitis.** (A) Representative images and (B) Quantification of immunohistochemistry staining for phosphorylated MEK1/2 (p-MEK1/2). Scale bars, 50 μm (upper panels) and 20 μm (lower panels). n = 4/group, $***p < 0.001$, Student's *t* test. (C) Representative images of immunofluorescence staining for F4/80+ (red) and p-MEK1/2+ (green) cells in femurs of *S. aureus* mice and controls, and (D) Quantification of the number of F4/80+ p-MEK1/2+ cells in right femurs infected with *S. aureus* and treated with vehicle from mice with *S. aureus* osteomyelitis and control, respectively. The number of F4/80+p-MEK1/2+ cells in the region around and away from abscess in right femurs of *S. aureus*-infected mice was quantified. Scale bars, 50 μm. n = 5/group. $*p < 0.05$, $***p < 0.001$, one-way ANOVA with Tukey's *post-hoc* test.

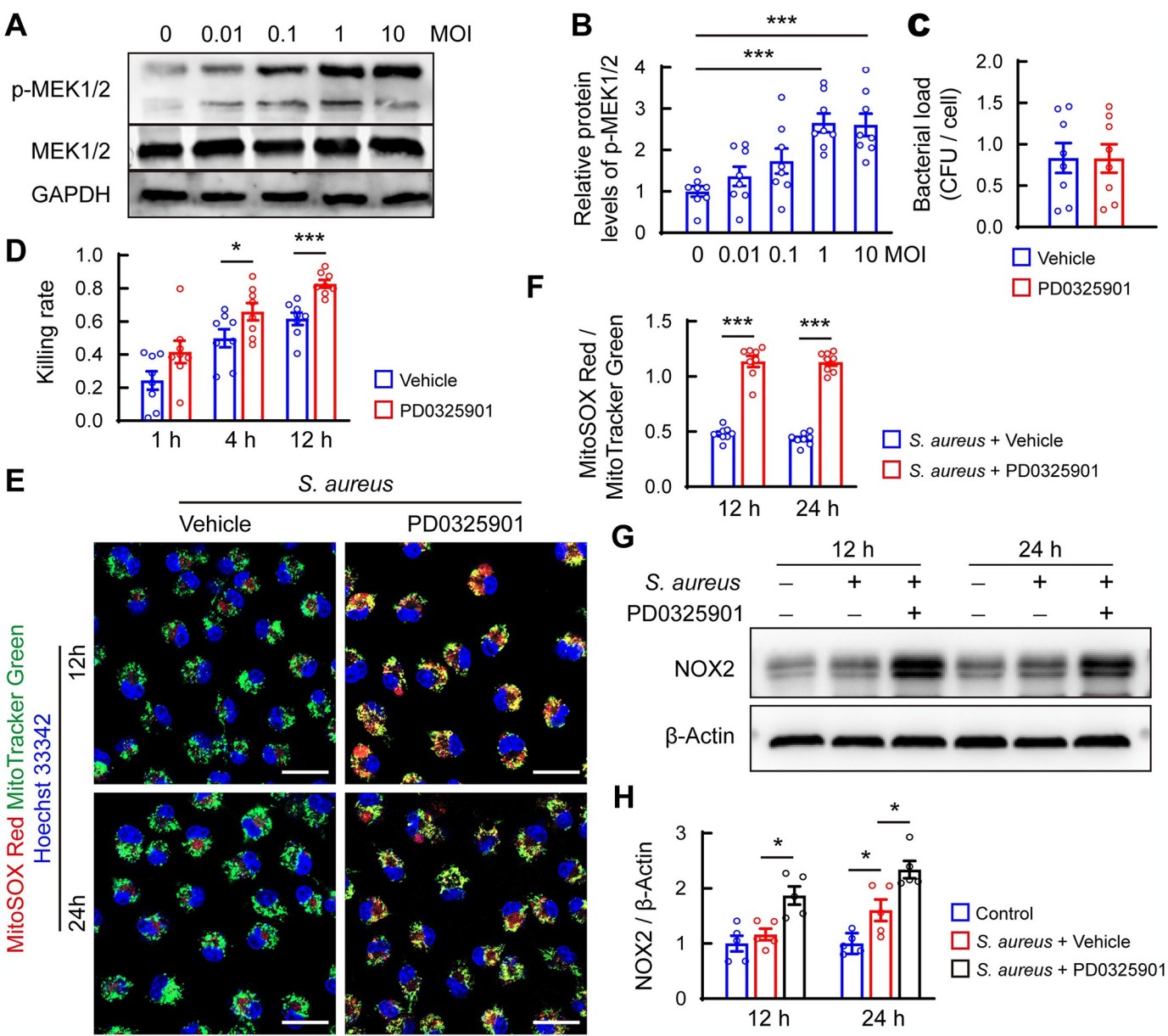

**Fig 3. Persistent *S. aureus* infection results in suppressed mtROS levels and impaired bactericidal ability by activating the MEK1/2 pathway in BMDMs.** (A) Representative images and (B) Quantification of western blots for phosphorylated MEK1/2 (p-MEK1/2) and total MEK1/2 levels in BMDMs. Cells were infected with *S. aureus* at a series dilution of multiplicity of infection (MOI, 0.01, 0.1, 1, and 10) for 30 min. n = 8/group, $^*$ $p < 0.05$, $^{***}p<0.01$ versus control. One-way ANOVA with Tukey's *post-hoc* test was used. (C) Quantification of *S. aureus* colonies for phagocytosis assay. BMDMs were pretreated with 1 μM PD0325901 or vehicle (DMSO) for 1 h, followed by *S. aureus* infection at MOI of 10 for 1 h. n = 8/group. (D) Quantification of *S. aureus* colonies for the bactericidal assay. BMDMs were infected with *S. aureus* at MOI of 10 for 1h. After removing non-phagocytosed extracellular bacteria, cells were treated with 1 μM PD0325901 or vehicle (DMSO), collected, and lysed at the indicated time points. n = 8/group. $^*$ $p < 0.05$, $^{***}$ $p < 0.001$, Student's *t* test. (E) Representative images of MitoTracker Green and MitoSOX Red fluorescence in *S. aureus*-infected BMDMs in the presence of 1 μM PD0325901 or vehicle (DMSO) for 12 h and 24 h. Scale bars, 10 μm. (F) Quantification of relative level of MitoSOX Red to MitoTracker Green. n = 8/group. $^*$ $p < 0.05$, $^{**}$ $p < 0.01$, Student's *t* test. (G) Representative images of western blot for NOX2 and (H) quantification of NOX2 levels normalized to β-Actin. n = 5/group. $^*$ $p < 0.05$, one-way ANOVA with Tukey's *post-hoc* test was used.

production but also disrupt phagocytic ROS production in BMDMs under prolonged *S. aureus* infection.

## EGFR signaling is essential for activating the MEK1/2 pathway in *S. aureus*-infected macrophages

Macrophages initiate the innate immune response by recognizing pathogens via pattern recognition receptors (PRRs), including Toll-like receptors (TLRs), nucleotide oligomerization domain (NOD)-like receptors (NLRs), and retinoic acid-inducible gene-I-like receptors (RLRs) [26]. To identify which PRR mediates MEK1/2 pathway activation in response to *S. aureus* infection, we used a series of specific inhibitors: ML130 for NOD1, GSK717 for NOD2, GSK2983559 for RIPK2, NOD-IN-1 for both NOD1 and NOD2, CU-CPT9a for TLR8, E6446 for TLR7 and TLR9, and C29 for TLR2. Our results showed that inhibition of TLR2 by C29 significantly blocked the activation of p-MEK1/2 by *S. aureus* infection, whereas TLR7/8/9 and NOD1/2 were not required for *S. aureus* to activate the MEK1/2 pathway in macrophages (S3A and S3B Fig).

To understand the molecular mechanism by which *S. aureus* activates the MEK1/2 pathway, we analyzed transcriptome data from femurs of mice with *S. aureus*-induced osteomyelitis and control samples on day 14 post-infection (GEO: GSE166522) from our previous work [27]. We overlapped MAPK cascade-related differentially expressed genes (DEGs) and DNA repair-related DEGs in Gene Ontology (GO) enrichment, and MAPK signaling pathway-related DEGs in Kyoto Encyclopedia of Genes and Genomes (KEGG) enrichment using the Venn diagram. Results identified EGFR as a central candidate DEGs (Fig 4A). We then assessed EGFR signaling activity in BMDMs in response to *S. aureus* infection. Results showed that the phosphorylation levels of EGFR were significantly upregulated by *S. aureus* at MOI of 1 and 10 (Fig 4B and 4C), with activation peaking at 30 min (S4A and S4B Fig).

Observing that *S. aureus* activated both EGFR signaling and the MEK1/2 pathway, we hypothesized that *S. aureus* might activate the MEK1/2 pathway through EGFR signaling. Indeed, treating BMDMs with erlotinib, an EGFR inhibitor, partially abolished MEK1/2 phosphorylation induced by *S. aureus* (Fig 4D and 4E). We then determined whether overactivation of EGFR signaling was associated with macrophage dysfunction. Erlotinib treatment did not affect the phagocytic ability of BMDMs (Figs 4F and S4C), but notably enhanced the bacterial killing rate of cells after 12 h of infection (Figs 4G and S4D). Consistent with functional rescue, erlotinib treatment also restored mtROS to higher levels after 12 h and 24 h of *S. aureus* infection (S4E and S4F Fig).

Next, we evaluated whether combining erlotinib and PD0325901 might further enhance the bactericidal function of BMDMs after 12 h of infection. Unexpectedly, we did not observe a synergistic effect (Figs 4H and S4G). Additionally, MitoSOX staining showed that both PD0325901 and erlotinib treatment upregulated mtROS levels, but their combination did not further increase mtROS levels after 12 h of *S. aureus* infection (Fig 4I and 4J). Thus, persistent *S. aureus* infection may suppress mtROS production, thereby restricting the bactericidal function of macrophages through the EGFR-MEK1/2-dependent pathway.

While mtROS is essential for fighting infection, uncontrolled production can trigger excessive inflammation [28,29]. We found that blocking the MEK1/2 pathway did not block the mRNA expression of IL-1β, IL-6, or TNF-α induced by *S. aureus* challenge, but strikingly upregulated IL-6 and TNF-α mRNA levels compared with vehicle-treated macrophages (S5A–S5C Fig). Interestingly, inhibition of EGFR signaling by erlotinib further increased IL-6 mRNA expression but markedly blocked IL-1β mRNA expression evoked by *S. aureus* infection (S5A–S5C Fig). These data indicate that EGFR signaling and the MEK1/2 pathway may

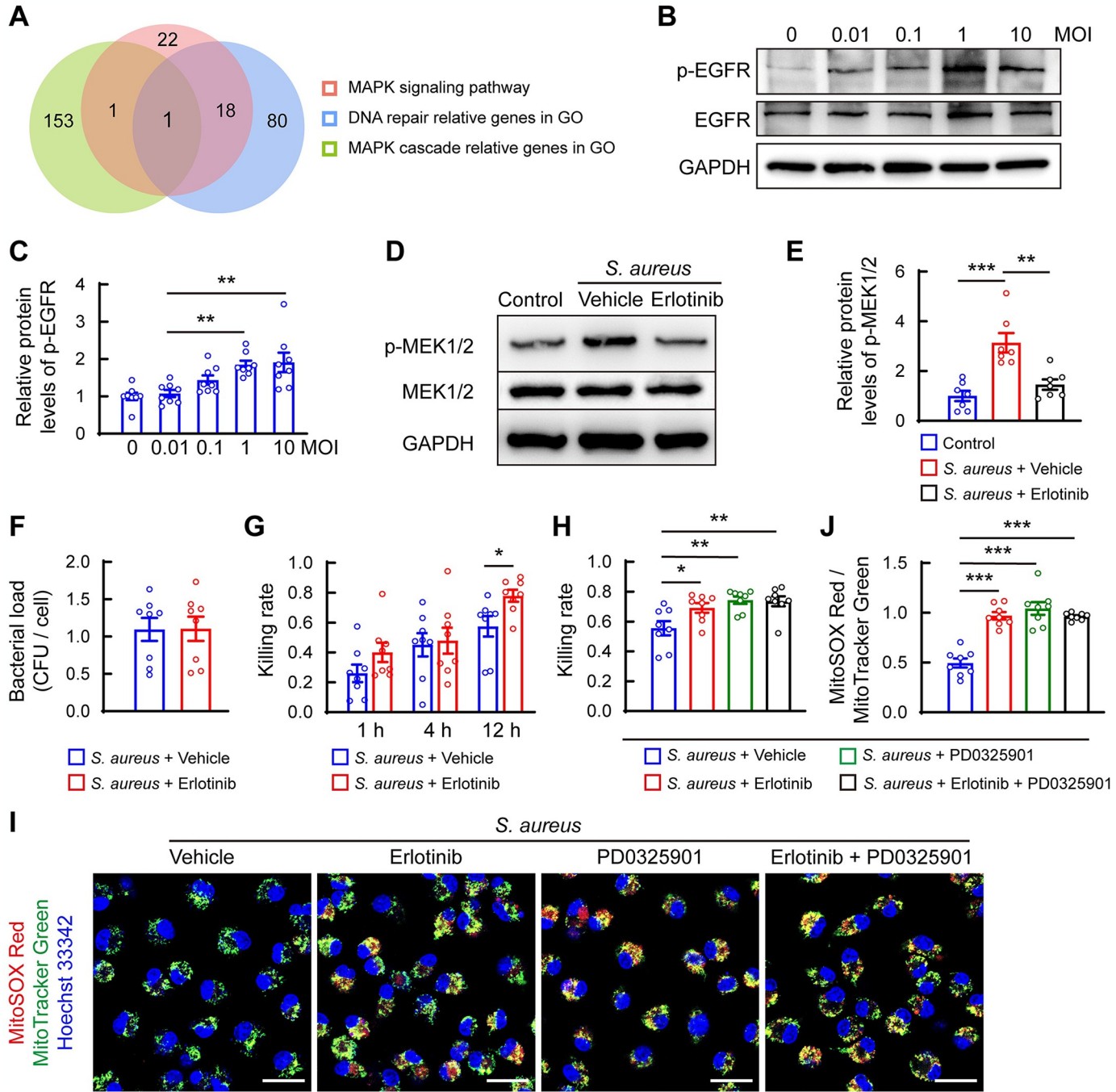

**Fig 4. *S. aureus* activates the MEK1/2 pathway by enhancing EGFR signaling in BMDMs.** (A) Venn diagram of DEGs associated with MAPK cascade and DNA repair in GO enrichment, and MAPK cascade in KEGG analysis. (B) Representative images of western blots and (C) Quantification of the ratio of phosphorylated EGFR (p-EGFR) to GAPDH in BMDMs after *S. aureus* infection (at the indicated MOI) for 30min. n = 8/group. ** $p < 0.01$ versus control, one-way ANOVA with Tukey's *post-hoc* test was used. (D) Representative western blot images of p-MEK1/2 and MEK1/2 and (G) Quantification of the ratio of p-MEK1/2 to total MEK1/2 in BMDMs. Cells were pre-treated with 10 μM erlotinib or vehicle (DMSO) for 1 h, following by *S. aureus* infection for 30 min. n = 7/group, ** $p < 0.01$, *** $p < 0.001$ versus control, one-way ANOVA with Tukey's *post-hoc* test was used. (F) Quantification of *S. aureus* colonies for phagocytosis assay. After BMDMs were pretreated with 10 μM erlotinib or vehicle (DMSO) for 1 h, cells were infected with *S. aureus* (MOI = 10) for 1 h. n = 8/group. (G) Quantification of intracellular colonies of *S. aureus* for the bactericidal assay. BMDMs were infected with *S. aureus* at MOI of 10 for 1h. After removing non-phagocytosed extracellular bacteria, cells were treated with 10 μM erlotinib or vehicle (DMSO), collected, and lysed at the indicated time points. n = 8/group, * $p < 0.05$, Mann-Whitney test. (H) Quantification of intracellular colonies of *S. aureus* for the bactericidal assay. BMDMs were infected with *S. aureus* at MOI of 10 for 1 h. After removing non-phagocytosed extracellular bacteria, cells were treated with 10 μM erlotinib, 1 μM PD0325901, a combination of 10 μM erlotinib and 1 μM PD0325901, or vehicle (DMSO) for 12 h. n = 8/group. * $p < 0.05$, ** $p < 0.01$, one-way ANOVA with Dunnett's *post-hoc* test was used. (I) Representative images and (J) Quantification of

mtROS in BMDMs detected using MitoTracker Green and MitoSOX Red. Cells were infected with *S. aureus* (MOI = 10) for 1 h. After removing non-phagocytosed extracellular bacteria, cells were treated with 10 μM erlotinib, 1 μM PD0325901, a combination of 10 μM erlotinib and 1 μM PD0325901, or or vehicle (DMSO) for 12 h. Scale bars, 10 μm. n = 8/group. *** $p < 0.001$, one-way ANOVA with Tukey's *post-hoc* test.

play differential roles in regulating inflammatory factor production in response to *S. aureus* infection.

Considering that TLR2 is essential for optimal innate immune defense against *S. aureus* infection [30], we evaluated the effect of C29, a TLR2 inhibitor, on EGFR-MEK1/2 cascade activity. Western blot results showed that C29 treatment significantly blocked MEK1/2 phosphorylation as early as 15 min after *S. aureus* challenge but had no effect on EGFR activation (Fig 5A–5C). These data indicate that TLR2 signaling contributes to MEK1/2 pathway activation but is not responsible for EGFR signaling activation in response to *S. aureus* challenge. Furthermore, phagocytosis assays revealed that C29 treatment did not affect the phagocytotic activity of BMDMs (Figs 5D and S6A). However, the bacterial killing assay showed that C29 treatment did not significantly affect bactericidal function at earlier time points (1 h and 4 h), but decreased bacterial killing rate and increased bacterial burden in BMDMs after 12 h of infection (Figs 5E and S6B), indicating that TLR2 signaling is critical for bacterial killing activity of macrophages after persistent *S. aureus* infection. Consistent with this finding, C29 treatment further decreased mtROS levels in BMDMs after 12 h of *S. aureus* infection (Fig 5F and 5G). These data demonstrate that the EGFR-MEK1/2 cascade might play a negative feedback role in TLR2 signaling-mediated mtROS production and bacterial killing during *S. aureus* infection.

## EGFR-MEK1/2 activation suppresses mtROS by inhibiting Chek2 expression

To further discover how EGFR-MEK1/2 activation suppresses mtROS production upon *S. aureus* challenge, we treated BMDMs with *S. aureus* for 12 h for transcriptome analysis. Analysis of DEGs associated with cellular responses to environmental stimuli and regulation of DNA damage checkpoints showed that three genes were significantly downregulated (Fig 6A and 6B). Chek2, a central effector of DNA damage response, mediates DNA checkpoint activation under stress conditions [31]. We confirmed that *S. aureus* infection suppressed Chek2 mRNA expression in BMDMs as early as 4 h post-infection (Fig 6C). The suppressed effect on Chek2 mRNA expression was alleviated by either PD0325901 or erlotinib treatment (Fig 6D). Immunofluorescence staining showed strikingly reduced levels of total and subcellular Chek2 in BMDMs after 12 h of *S. aureus* infection (Fig 6E–6H). Notably, both PD0325901 and erlotinib treatments significantly increased Chek2 intensity in the cytoplasm and nucleus (Fig 6E–6G), with enhanced Chek2 localization in mitochondria (Fig 6H). Consistent with the effect on bactericidal function, combined treatment with PD0325901 and erlotinib did not further increase Chek2 protein levels (Fig 6E–6H). Considering that Chek2 activation contributes to high mtROS production [32], we evaluated the function of Chek2 in mtROS production in response to *S. aureus* infection. Our results showed that inhibition of Chek2 by BML-277 blocked mtROS activation induced by PD0325901 or erlotinib treatment (Figs 6I and 6J). Taken together, these results demonstrate that EGFR and MEK1/2 activation by *S. aureus* infection reduces mtROS levels by suppressing Chek2 expression, thereby impairing macrophages' bactericidal function.

Next, we investigated whether blocking EGFR signaling could reduce bacterial load and rescue bone destruction *in vivo*. Mice with implant associated osteomyelitis were treated with

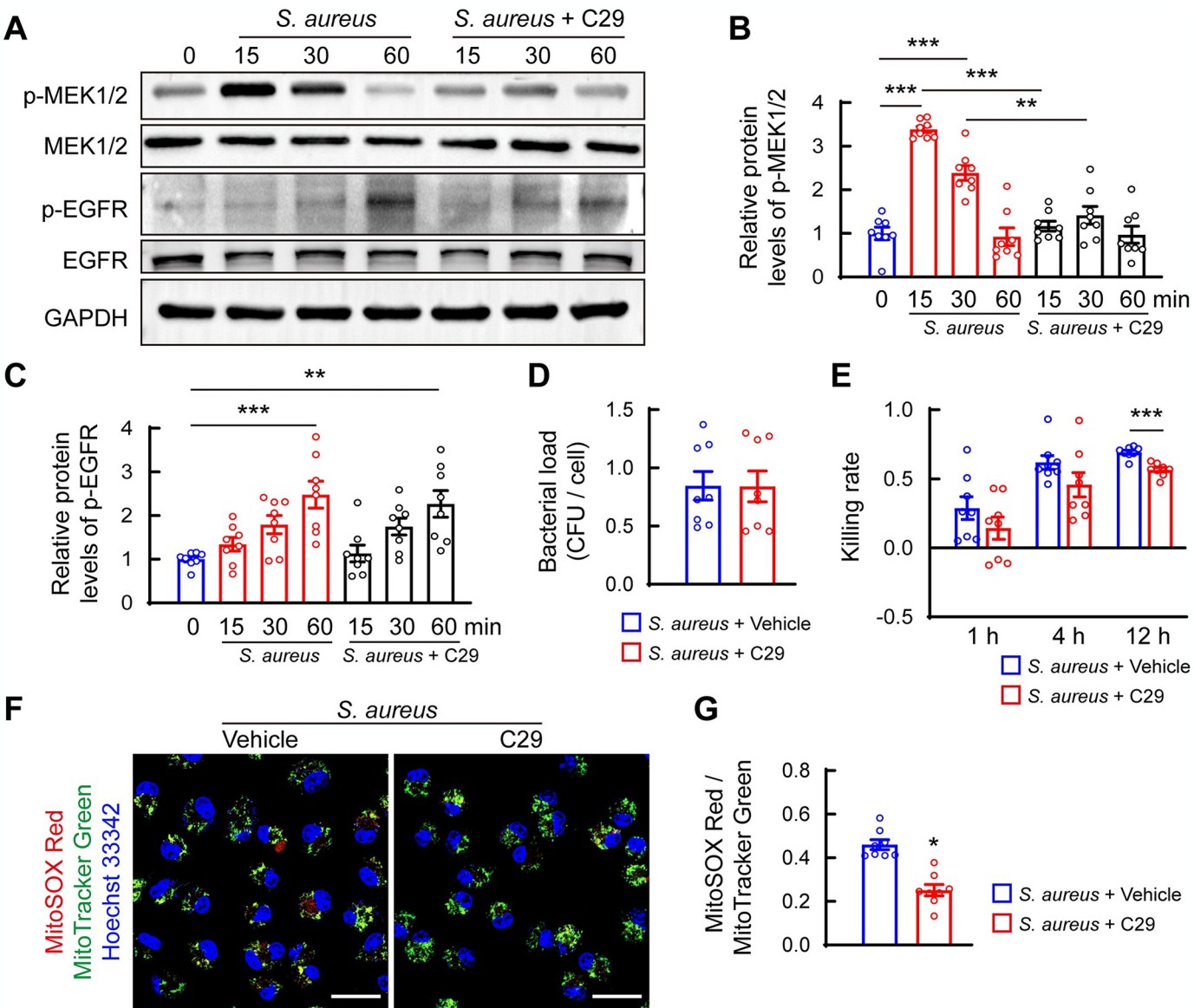

**Fig 5. TLR2 signaling is not responsible for EGFR-MEK1/2 pathway-mediated bactericidal function in BMDMs.** (A) Representative images and quantification of western blot for p-MEK1/2 and total MEK1/2 (B), p-EGFR and total EGFR (C) levels in BMDMs. After pretreatment with 100 μM C29 or vehicle (DMSO) for 1 h, cells were infected with *S. aureus* at MOI of 10, and whole cell lysates were collected at the indicated time points for western blot analysis. $^{**}p < 0.01$, $^{***}p < 0.001$, one-way ANOVA with Tukey's *post-hoc* test. (D) Quantification of intracellular colonies of *S. aureus* for phagocytosis assay. BMDMs were pretreated with 100 μM C29 or vehicle (DMSO) for 1 h, followed by *S. aureus* infection at a MOI of 10 for 1 h. n = 8/group. (E) Quantification of intracellular colonies of *S. aureus* for the bacterial killing assay. After removing the extracellular bacteria, BMDMs were cultured in growth media with presence or absence of 100 μM C29 for 12 h. n = 8/group, $^{***}\ p < 0.001$, Student's *t* test. (F) Representative images and (G) Quantification of mitochondrial ROS (mtROS) levels in BMDMs. n = 8/group, $^*\ p < 0.05$, Student's *t* test. After removal of the extracellular *S. aureus* and additional 12 h of culture with 100 μM C29 or vehicle (DMSO), mtROS levels were detected using MitoTracker Green and MitoSOX Red. Nuclei were stained with Hoechst 33342.

erlotinib or vehicle. Immunofluorescence staining revealed upregulation of p-EGFR levels in F4/80[+] macrophages in the bone marrow of *S. aureus*-infected femurs of mice treated with vehicle, whereas erlotinib treatment suppressed it (S7A Fig). Importantly, erlotinib-treated mice had significantly reduced bacterial load in *S. aureus*-infected femurs compared with vehicle-treated mice (Fig 7A and 7B). Micro-CT imaging showed that erlotinib treatment

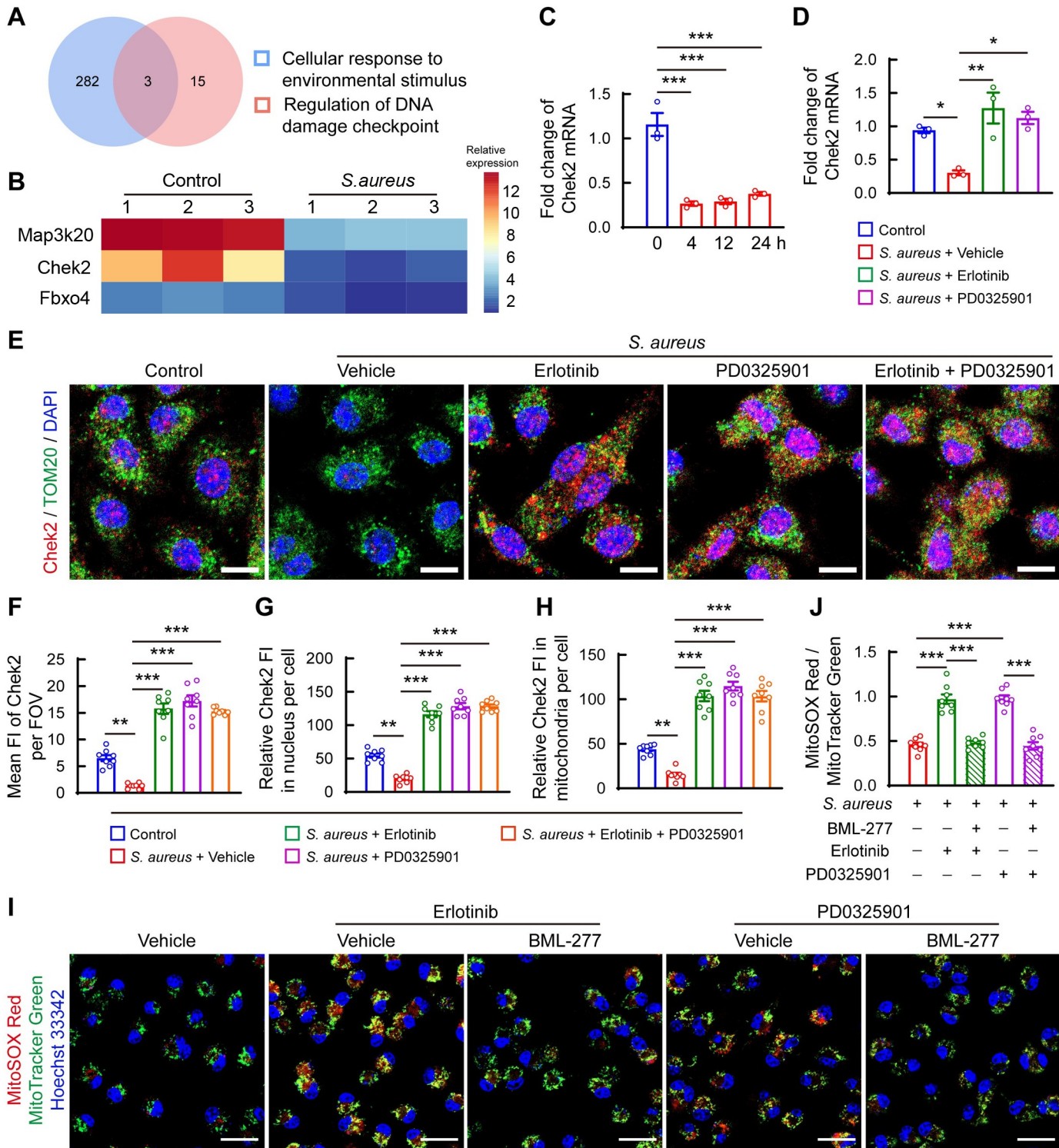

**Fig 6. EGFR-MEK1/2 cascade suppresses mtROS levels by inhibiting Chek2 expression in BMDMs.** (A) Venn diagram of DEGs associated with cellular response to environmental stimuli and regulation of DNA damage checkpoints. (B) Heatmap of overlapped DEGs in the Venn diagram. (C) mRNA expression of checkpoint kinase 2 (Chek2) in BMDMs. Cells were infected with *S. aureus* (MOI = 10) for 1 h. After removal of extracellular bacteria, cells were cultured in growth media for the indicated time points. n = 3/group, *** $p < 0.001$ versus control, one-way ANOVA with Tukey's *post-hoc* test. (D) mRNA expression of Chek2 in BMDMs. After phagocytosis for 1 h and removal of the extracellular bacteria, cells were treated with 10 μM erlotinib, 1 μM PD0325901 or vehicle (DMSO) for 12 h. n = 3/group, * $p < 0.05$, ** $p < 0.01$, one-way ANOVA with Tukey's *post-hoc* test. (E) Representative confocal images of immunofluorescence staining for Chek2 (red) and TOM20-labeld mitochondria (green). Scale bars, 10 μm. Nuclei were stained with DAPI. (F-H) Quantification of mean fluorescence intensity (FI) of Chek2 in the

field of view (FOV) (F), the FI of Chek2 in the nucleus (G) and mitochondria (H) per cell. n = 8/group, **$p < 0.01$, *** $p < 0.001$, one-way ANOVA with Dunnett's *post-hoc* test. (I) Representative confocal images and (J) Quantification of mitochondrial ROS (mtROS) levels detected using MitoTracker Green and MitoSOX Red in BMDMs. Nuclei were stained with Hoechst 33342. After phagocytosis for 1 h and removal of the extracellular *S. aureus*, cells were treated with 5 μM BML-277 or vehicle (DMSO) with or without 1 μM PD0325901 or 10 μM erlotinib. mtROS levels were detected after 12 h of treatment. Scale bars, 20 μm. ***$p < 0.01$, one-way ANOVA with Tukey's *post-hoc* test.

significantly reduced reactive new bone formation around cortical bone and decreased cortical bone loss (Fig 7C–7E). Consistently, erlotinib-treated mice showed notably increased bone mineral density (BMD) and bone volume to total volume (BV/TV), primarily due to rescued trabecular bone number (Tb.N) (Fig 7C and 7F–7H). However, there were no changes in trabecular bone thickness (Tb.Th) and trabecular pattern factor (Tb.Pf) in erlotinib-treated mice compared with vehicle-treated mice (Fig 7I and 7J). Histological staining and histopathological score analysis confirmed the positive effect of erlotinib treatment on bone structure in mice with *S. aureus* osteomyelitis (S7B and S7C Fig). Specifically, vehicle-treated mice showed deformed femurs with extensive abscess formation in the bone cavity and loss of trabecular bone in the metaphyseal area, whereas erlotinib-treated mice had improved trabecular bone structure with limited abscess formation around the implant (S7A Fig). These data indicate that blocking EGFR signaling alleviates bacterial burden and improves bone structure.

To confirm the role of EGFR signaling in the MEK1/2 pathway, we evaluated its expression in vehicle- and erlotinib-treated mice with *S. aureus* osteomyelitis. Immunofluorescence staining showed that erlotinib treatment substantially decreased p-MEK1/2 levels in F4/80[+] cells (Fig 7K and 7L). We then evaluated whether inhibiting EGFR signaling could rescue Chek2 expression in macrophages. Results showed that erlotinib treatment markedly restored decreased Chek2 expression in *S. aureus*-infected femurs (Fig 7M and 7N). Together, our findings indicate that *S. aureus* infection may activate EGFR-MEK1/2 signaling, thereby suppressing mitochondrial ROS levels by downregulating Chek2 expression, leading to the progression of *S. aureus* osteomyelitis in mice.

## Discussion

Macrophages are essential components of the first line of host defense against *S. aureus* infection. However, *S. aureus* employs multiple self-defensive strategies to survive within and escape macrophages, the mechanism of which remains largely unexplored. Here, we demonstrate that blocking the EGFR-MEK1/2 pathway may be a promising strategy to strengthen host defense against *S. aureus* osteomyelitis. Importantly, we uncover a previously unknown aspect of the EGFR-MEK1/2 cascade involved in intracellular survival of *S. aureus* in macrophages: suppression of mtROS production through downregulation of Chek2 expression, which reduces cell bactericidal activity. This study therefore identifies the EGFR-MEK1/2 cascade in macrophages as a critical target for the treatment of *S. aureus* osteomyelitis.

Recent studies have highlighted the important role of the MEK1/2 pathway in sustaining the replication of *Salmonella* and viruses in cells [33–35], emphasizing the critical function of MEK1/2 overactivation in infection propagation. Similarly, our current work further reveals the critical role of the MEK1/2 pathway in suppressing the bactericidal activity of macrophages. We show that overactivation of the MEK1/2 pathway leads to reduced mtROS levels, impairing the bactericidal function of macrophages. Moreover, blocking the MEK1/2 pathway reduces bacterial load and mitigates bone destruction in mice with osteomyelitis, suggesting that targeting the MEK1/2 pathway may be a potential therapeutic approach to restore compromised immunity in chronic *S. aureus* osteomyelitis. It is important to note that abnormal activation of this pathway has also been linked to inflammatory tissue lesions, and blocking

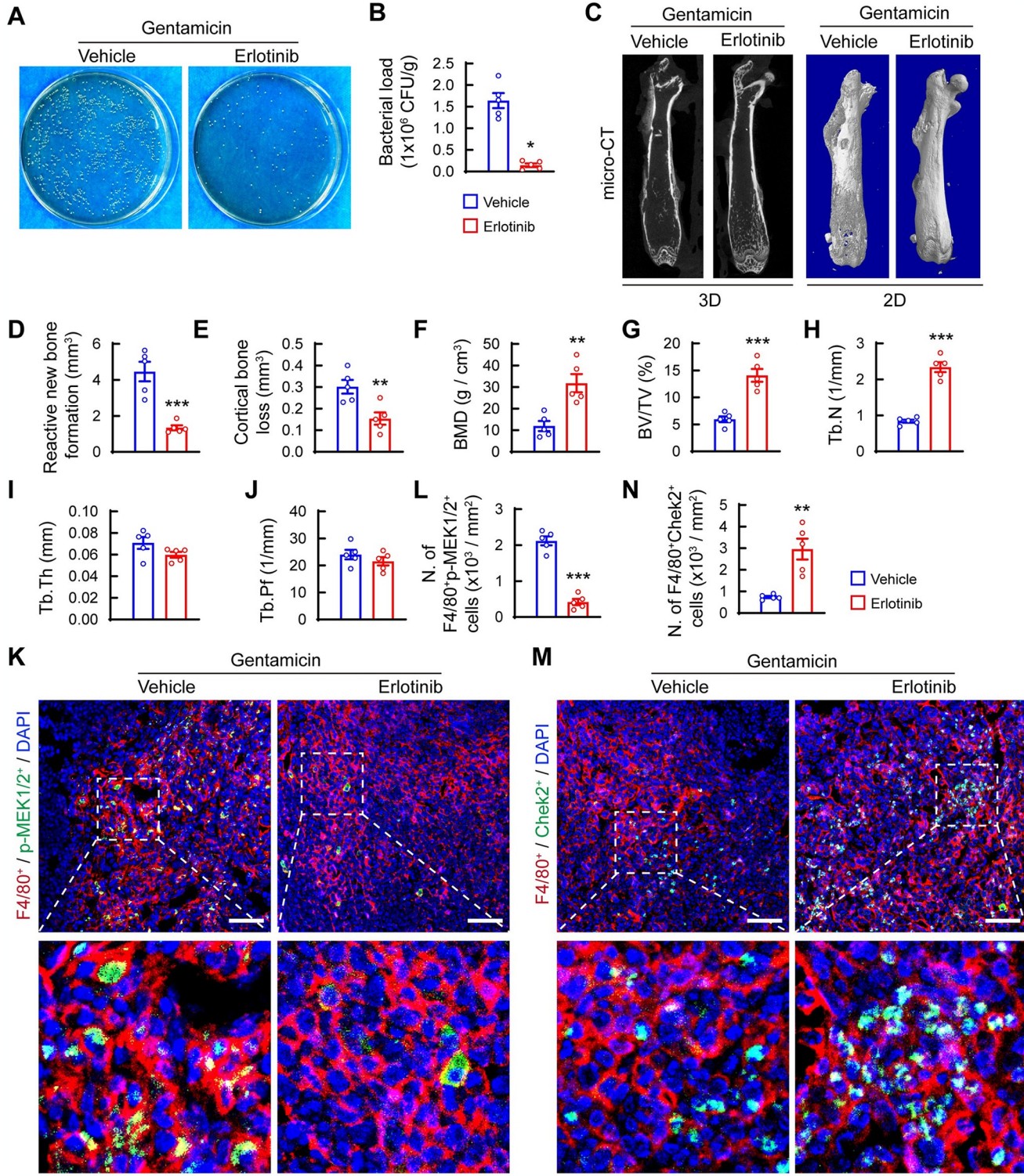

**Fig 7. Blocking EGFR signaling reduces bacterial loading and bone microstructure in femurs infected with *S. aureus*.** (A) Representative images and (B) Quantification of *S. aureus* colonies growing on an agar plate after recovery from infected femurs. (C) Representative 3D and 2D coronal micro-CT images and quantification of reactive new bone formation (D), cortical bone loss (E), bone mineral density (BMD) (F), bone fraction (BV/TV) (G), trabecular thickness (Tb.Th) (H), trabecular number (Tb.N) (I), and trabecular bone pattern factor (Tb.Pf) (J). n = 5/group. (I) Representative images and (J) Quantification of immunofluorescence staining for F4/80+ (red) and p-MEK1/2+ (green) cells in *S. aureus*-infected femurs of mice treated with erlotinib and vehicle. Scale bars, 50μm. n = 5/group. (K) Representative images and (L) Quantitative analysis of F4/80+ (red) and p-MEK1/2+ (green) cells in *S.*

*aureus*-infected femurs of mice treated with erlotinib and vehicle. Scale bars, 50μm. n = 5/group. (M) Representative images and (N) Quantitative analysis of F4/80$^+$ (red) and Chek2$^+$ (green) cells in *S. aureus*-infected femurs of mice treated with erlotinib and vehicle. Scale bars, 50μm. n = 5/group. $^{**}$ $p < 0.01$, $^{***}$ $p < 0.001$. Statistical significance was analyzed by Student's $t$ test.

MEK1/2 is believed to alleviate pro-inflammatory responses [17,35,36]. Contrary to this belief, we observed that blocking the MEK1/2 pathway led to further upregulation of inflammatory factors in *S. aureus*-infected BMDMs *in vitro*. Therefore, the notable improvement in bone structure by blocking the MEK1/2 pathway in the femurs of mice with *S. aureus* osteomyelitis is likely due to the enhanced bactericidal activity of macrophages in the bone.

Our finding that EGFR signaling is activated by *S. aureus* in BMDMs is both surprising and interesting. EGFR, a transmembrane protein with tyrosine kinase activity, is ubiquitously expressed in various cells and aberrantly activated in lung cancer, colorectal cancer, and other diseases [37]. Increasing evidence supports the pivotal roles of EGFR signaling in pathogenic bacterial infection, such as invasion of *Serratia proteamaculans* into m-Hela cells [38], induction of inflammation in epithelia cells in response to *Candida albicans* infection [39], and activation of inflammatory responses in macrophages by *Helicobacter pylori* infection [40]. Our current findings reveal a novel role of EGFR signaling upon *S. aureus* infection: its overactivation is associated with reduced bacterial killing activities in macrophages. Additionally, our studies demonstrate that activation of EGFR signaling by *S. aureus* may lead to downregulation of mtROS levels and reduction of macrophage bactericidal function via the MEK1/2 pathway. However, the MEK1/2 pathway is also a key mediator of TLRs signaling-mediated innate immune responses [41]. Supporting this, our data indicate that TLR2 signaling is required to activate the MEK1/2 pathway in the context of *S. aureus* infection. Although studies have reported mutual activation between EGFR and TLR signaling in the presence of microbe [42,43], we have not observed inhibition of EGFR signaling by blocking TLR2 during *S. aureus* challenge. Therefore, it is reasonable to presume that EGFR and TLR2 signaling may independently regulate the MEK1/2 pathway in response to *S. aureus* infection.

Accumulating evidence indicates that mtROS is a critical component of antimicrobial responses in macrophages [22,44,45]. Supporting this, our previous study showed that prolonged infection (12 h or longer) leads to a dramatic decline in mtROS levels and reduced bactericidal activity in macrophages [23]. Our current data further identify Chek2, a positive regulator of mtROS production [29], is negatively regulated by the EGFR-MEK1/2 cascade during *S. aureus* infection. Despite the pivotal role of mtROS in the antimicrobial functions of innate immune cells, a strong mtROS signal is critical for the proinflammatory response, which can damage cells and tissues during infection [46–48]. Our data reveal that EGFR signaling and MEK1/2 may collaboratively regulate mtROS levels and bactericidal function of macrophages via Chek2 while differentially control inflammatory responses upon *S. aureus* infection. It is important to note that while blocking TLR2 with C29 significantly suppresses p-MEK1/2 levels, it also decreases mtROS production and impairs the bactericidal function of macrophages during persistent *S. aureus* infection. This indicates that TLR2 is crucial for maintaining macrophage antimicrobial function through pathways other than MEK1/2.

## Conclusion

The current study has extended our understanding of the critical role of EGFR-MEK1/2 cascade in the persistent infection of *S. aureus* in macrophages. By blocking the MEK1/2 pathway both *in vivo* and *in vitro*, we identified MEK1/2 as a hub pathway in negatively regulating mtROS levels and compromising the bactericidal function of bone marrow macrophages

during *S. aureus* infection. Although the precise mechanism by which the MEK1/2 pathway inhibits Chek2 expression and suppresses mtROS production requires further investigation, our data suggest that downregulation of Chek2 expression in cells contributes to reduced mtROS levels and subsequently suppressed bactericidal function of macrophages. Taken together, our findings indicate that over activation of the EGFR-MEK1/2 cascade may be a core event in suppressing bactericidal function during *S. aureus* infection. Therefore, targeting this pathway may represent a promising avenue for breakthrough therapies for *S. aureus* osteomyelitis.

## Materials and methods

### Bacteria culture

The *S. aureus* used in this study was isolated from an osteomyelitis patient, as previously described [27]. *S. aureus* strains were maintained at -80˚C in tryptic soy broth (TSB) containing 25% (v/v) glycerol. They were cultured from cold storage by plating on tryptic soy agar (TSA) at 37˚C. A single clone of *S. aureus* was selected and cultured in TSB by shaking at 200 rpm for 16–18 h before being collected by centrifugation at 2500g for use *in vitro* and *in vivo* infection experiments. After resuspension in phosphate buffered saline (PBS), *S. aureus* strains were adjusted to $1 \times 10^8$ CFU/ml by measuring optical density at 600nm ($OD_{600}$) of 0.5.

### Implant-associated *S. aureus* osteomyelitis mouse model and treatments

All mouse care and treatment procedures followed guidelines approved by the Animal Care and Use Committee of Southern Medical University Nanfang Hospital. Male C57BL/6 mice (8–10 weeks) were purchased from Southern Medical University Animal Center (Guangzhou, China). All mice were housed under standard conditions (23 ± 2˚C, 12 hours light-dark cycle) with free access to water and food.

The mouse model of implant-associated *S. aureus* osteomyelitis was established as previously described [27]. Briefly, male C57BL/6 mice aged 10–12 weeks were anaesthetized with tribromoethanol (250 mg/kg). After exposing the right femur by blunt dissection, a small hole was drilled in the mid-shaft with a 27-gauge needle. A sterilized stainless pin, 2 mm in length, was inserted through the hole, followed by slow injection of 2 μl *S. aureus* suspension ($1 \times 10^5$ CFU/ml) or PBS (control group) into the intramedullary canal. The hole was seamed with bone wax, the incision was sutured and disinfected.

To evaluate the effect of MEK1/2 blocking on the progression of *S. aureus* osteomyelitis in mice, *S. aureus*-infected mice were randomly assigned to treatment with PD0325901 (0.2 mg/kg/day, i.p., #S1036, Selleck) or vehicle (5% dimethyl sulfoxide (DMSO) in normal saline) from day 5 post-infection. To assess the role of EGFR signaling in the pathogenesis of *S. aureus* osteomyelitis in mice, *S. aureus*-infected mice were randomly treated with erlotinib (10 mg/kg/day, i.p., #HY-1208, MCE) or vehicle (10% DMSO in corn oil) from day 5 post-infection. Gentamicin at 10 mg/kg had a poor killing effect on intracellular *S. aureus* in a mouse model of peritonitis [49], and our recent work showed that a substantial amount of *S. aureus* resides in *Lyz2*[+] macrophages in the bone marrow of osteomyelitis mice treated with gentamicin at a dose of 20 mg/kg/d [50]. To investigate the role and mechanism of intracellular *S. aureus* in pathogenesis of osteomyelitis, all mice were treated with gentamicin (20 mg/kg/day, i.p.) from day 1 post-surgery to reduce the extracellular burden of *S. aureus* in infected femurs. The mice were euthanized on day 14 after surgery, and their implanted right femurs were collected for further analysis.

## Immunohistochemistry and immunofluorescence

For immunohistochemistry staining, femurs were fixed in 4% paraformaldehyde overnight, decalcified in 10% ethylenediamenetetraacetic acid (EDTA) solution for 10 days, and processed for paraffin-embedding. Four-μm-thick sections were placed on glass slide, deparaffinized, and rehydrated before antigen retrieval. Endogenous peroxidase activity was blocked by incubating sections in a 0.3% (v/v) hydrogen peroxide solution for 15 min. After blocking with goat serum for 1 h at room temperature, sections were incubated with phospho-MEK1/2 antibody (AF3385, Affinity, China) overnight at 4˚C. Samples were then incubated in HRP-conjugated goat anti-rabbit secondary antibody (HA1001, Huabio, China) for 1 h. Peroxidase activity was revealed using a 3,3'-diaminobenzidine (DAB) kit (ZLI-9018, ZSGB-BIO, China) and nuclei were counterstained with hematoxylin (H-3404-100, Vector, USA). Image pro plus 6.0 was used to analyze the integrated optical density of p-MEK1/2 in the bone marrow.

For immunofluorescence staining, bone cyosections were thawed, washed with PBS, and permeabilized using 0.1% Triton-X100 in PBS (0.1% PBST) for 10 min. After incubation in blocking buffer (10% normal goat serum, 0.1% PBST), sections were incubated with primary antibodies in blocking buffer overnight at 4˚C. The primary antibodies used were rabbit anti-*S. aureus* antibody (Ab20920, Abcam, USA), rat anti-F4/80 antibody (MCA497GA, BIO-RAD, USA), rabbit anti-MEK1/2 (phosphor-S218/222) (BS4733, Bioworld, China), rabbit anti phosphor-EGFR (Tyr1068) (3777s, Cell signaling Technology, USA), and mouse anti-TOM20 monoclonal antibody (66777-1-lg, Proteintech, Wuhan, China). After being washed three times, sections were stained with Anti-rabbit IgG(H+L) Alexa Fluor 488 Conjugate (4412s, Cell Signaling Technology, USA), Goat anti-Rat IgG (H+L) Alexa Fluor 594 (ab150160, Abcam, USA), or goat anti-mouse IgG (H+L) Alexa Fluor 488 Conjugate (4408s, Cell Signaling Technology, USA) in blocking solution at room temperature for 1 h. Nuclei were counterstained with a 4′,6-diamidino-2-phenylindole (DAPI) solution (E607303-0002, BBI Life Science, Shanghai, China).

For immunofluorescence labeling in cell cultures, cells were washed with PBS three times before fixation in 4% paraformaldehyde for 10 min. After washing in PBS, cells were incubated in blocking buffer at room temperature for 1 h, followed by incubation with rabbit anti-checkpoint kinase 2 (Chek2) primary antibody (13954-1-AP, Proteintech, China) and mouse anti-TOM20 monoclonal antibody (66777-1-lg, Proteintech, Wuhan, China) overnight at 4˚C. The following day, after washing in 0.1% PBST, cells were incubated with CoraLite594-conjugated goat anti-rabbit IgG (H+L) (SA00013-4, Proteintech, China) and goat anti-mouse IgG (H+L) Alexa Fluor 488 Conjugate (4408s, Cell Signaling Technology, USA) for 2 h. DAPI was used to label nuclear DNA. Image Z-stacks were obtained using a fluorescence microscope (Zeiss, LSM980, Baden-Württemberg, Germany).

## Microcomputed tomography (micro-CT)

Bone destruction and new bone formation were assessed using micro-CT imaging with Skyscan (Bruker, New York, USA). Each femur was imaged at a pixel size of 9 μm, with a voltage of 70 kV, a current of 145 mA, and an integration time of 300 ms. The 3D reconstruction images were processed using Image Processing Language V5.15 software. To evaluate cortical bone destruction and reactive bone formation, the region of interest (ROI) was centered in the middle of the femur, where the pin was implanted. To analyze changes in trabecular bone microstructure in the distal femur, the lower border was set 0.5 mm away from the growth plate, and the upper border was defined as 2 mm from this position longitudinally. Parameters such as cortical bone loss, reactive new bone formation, bone mineral density (BMD), bone

volume/tissue volume (BV/TV), trabecular thickness (Tb.Th), trabecular number (Tb.N) and trabecular bone pattern factor (Tb.Pf) were measured using SkyScan1176 software.

## Histological analyses

Tibias and femurs were fixed in 4% paraformaldehyde for 48 h, decalcified in 10% EDTA solution for 10 days, and processed for paraffin-embedding. Hematoxylin-eosin (H&E) staining was performed following standard procedures. Histopathological signs of osteomyelitis were scored using the methods described by Smeltzer et al. [51], based on the presence of acute intraosseous inflammation, chronic intraosseous inflammation, periosteal inflammation, and bone necrosis, each graded on a scale of 0–4.

## Bacterial burdens in implanted-femurs

Fourteen days post-infection, mice were anaesthetized with tribromoethanol (250 mg/kg) and euthanized by cervical dislocation, implanted femurs were dissected free of soft tissue, and after removal of intramedullary implants, femurs were homogenized in 1 ml PBS. *S. aureus* burdens were determined by plating serial dilutions of homogenate on TSA plates. After 12 h of incubation at 37°C, bacterial colonies were imaged and counted using ImageJ software (V1.8.0, National Institute of Health).

## Isolation and culture of bone marrow-derived macrophages (BMDMs)

BMDMs were prepared from 8–10 weeks old C57BL/6 mice following established protocols [23]. Mice were euthanized by cervical dislocation, and bone marrow cells were flushed from the femora and tibiae. After lysis of red blood cells with ACK lysis buffer, bone marrow cells were cultured and differentiated in RPMI-1640 medium supplemented with 30% L929 conditioned medium, 10% fetal bovine serum, and 1% penicillin/streptomycin. Cells were seeded at a density of $2.0 \times 10^6$/ml in 12-well or 6-well plates for 7 days, with medium refresh every 2 days. On day 8 of differentiation, the cells were treated for further analysis.

## Western blotting

To evaluate upstream signals involved in activating MEK1/2 pathway of BMDMs in response to *S. aureus* challenge, cells were pretreated with 5 μM ML130, 5 μM GSK717, 0.5 μM GSK2983559, 10 μM NOD-IN-1, 5 μM CU-CPT9a, 1 μM E6446, or the same volume of vehicle (DMSO, final dilution ratio: 1/1000) for 1 h, followed by *S. aureus* (MOI = 10) infection. Control cells were treated with DMSO (1/1000) for 1 h, followed by PBS (vehicle for *S. aureus* suspension, 7.5% in medium) for 30 min. Cells were lysed with radio immunoprecipitation assay (RIPA) lysis buffer, and whole-cell lysates were harvested. Protein samples were separated on 10% SDS-polyacrylamide gels by electrophoresis, transferred to polyvinylidene difluoride membranes, and blocked with 5% skim milk. Blots were incubated with one of the following primary antibodies overnight at 4°C. The primary antibodies used in this study were rabbit anti-MEK1/2 (phosphor-S218/222) (ET1609, HUABIO, China), rabbit anti-MEK1/2 (8727T, CST, USA), rabbit anti-Phospho (Y1092)-EGFR (ET1606-44, Huabio, Hangzhou, China), mouse anti-p-Tyr (PY20) (sc-508, Santa Cruz, USA), rabbit anti-EGFR (ET1603-37, Huabio, Hangzhou, China), rabbit anti-NOX2 (19013-1-AP, Proteintech, Wuhan, China), rabbit anti-p22 (DF10099, Affinity, China), rabbit anti-p47 (AF5220, Affinity, China), rabbit anti-phospho-p47 (CSB-PA050089, CUSABIO, Wuhan, China), rabbit anti-p67 (15551-1-AP, Proteintech, Wuhan, China), rabbit anti-p40 (DF6820, Affinity, China), mouse anti-β-Actin (66009-1-lg, Proteintech, Wuhan, China), and rabbit anti-GAPDH (ET1601-4, Huabio, Hangzhou,

China). Next, blots were incubated with HRP-conjugated Goat anti-Rabbit secondary antibody (HA1001, Huabio, Hangzhou, China) or HRP-conjugated Goat anti-Mouse IgG (HA1006, Huabio, Hangzhou, China) for 1 hour at room temperature. Chemiluminescent signals were developed in Immubilon Western Chemilum HRP substrate (WBKLS0500, Merck Millipore, Germany), and detected with a BLT multifunctional imaging station (GelView 6000 Pro, Biolight Biotechnology Co. Guangzhou, China). Band intensities were quantified using ImageJ software (V1.8.0, National Institute of Health). The band intensities of each protein were normalized to GAPDH or β-Actin.

## Bacterial phagocytosis and killing capability

To measure phagocytosis, BMDMs were pretreated with 1 μM PD0325901 (S1036, Selleck, China), 10 μM erlotinib (HY-50896, MedChemExpress, China), 100 μM C29 (HY-100461, MedChemExpress, China), or the same volume of vehicle (DMSO, final dilution ratio: 1/1000) for 1 h. Subsequently, the BMDMs were incubated with *S. aureus* at multiplicity of infection (MOI) of 10 for an additional 1 h. After washing with PBS, non-phagocytosed bacteria were eliminated using 20 μg/mL lysostaphin and 50 μg/mL gentamicin. Cells were then lysed in 0.1% (v/v) Triton-100, and serial dilutions were plated on TSA plates to enumerate bacterial colonies.

For evaluating intracellular bacterial killing capacity, after the removal of extracellular bacteria, infected BMDMs were cultured in medium containing 1% penicillin & streptomycin for 1, 4 or 12 h, with or without 1 μM PD0325901, 10 μM erlotinib, or 100 μM C29. At the indicated time points, cells were lysed, and diluted aliquots were spread on TSA plates. Bacterial colonies were counted after overnight incubation at 37°C.

## Mitochondrial radical oxygen species (mtROS) analysis

Mitochondrial ROS levels were detected using MitoSOX Red (Invitrogen, Massachusetts, USA) and MitoTracker Green (Invitrogen, Massachusetts, USA), following manufacturer instructions. Hoechst33342 (Solarbio, Beijing, China) was used to label nuclear DNA. Images were captured from 4–6 randomly selected fields using confocal laser microscopy (Zeiss, LSM980, Baden-Württemberg, Germany). Data analysis was performed with Image J software (V1.8.0, National Institute of Health).

## RNA sequencing and data analysis

Our previous RNA sequencing data from femurs of mice with *S. aureus* osteomyelitis and controls, deposited in Gene Expression Omnibus (GEO) database (assessing number: GSE166522) were analyzed. Differentially expressed genes (DEGs) ($p$ value < 0.05, and |Log$_2$(fold change)|>1) were subjected to Gene Ontology (GO) and Kyoto Encyclopedia of Genes and Genomes (KEGG) analyses using clusterProfiler R package (Version 3.6.2). GO and KEGG terms with adjusted $p$ values of < 0.05 were considered significant enrichment.

For RNA sequencing in BMDMs, cells were infected with *S. aureus* at a MOI of 10 for 1 h, followed by treatment with 20 μg/mL lysostaphin and 50 μg/mL gentamicin to remove extracellular bacteria. After three washes with PBS, BMDMs were cultured for an additional 24 h. Afterwards, cells were washed with PBS, lysed in Trizol (9108, Takara, Japan), and sent to OmicShare (Guangzhou, China) for next-generation sequencing. RNA was quality-assessed with Agilent 2100 (Agilent, USA) using RNA 6000 Nano kit (5067–1511, Agilent, USA), with RNA integrity number above 9 for library construction. RNA samples from three independent experiments were used for library construction following the protocol of Illumina NovaSeq 6000 platform and sequenced using a Thermal Cycler (Eastwin, Suzhou, China) with

HighSensitivity DNA assay Kit (5067–4626, Agilent, USA). FASTq sequencing files were aligned to the mouse reference genome (GRCm38, mm10) using STAR aligner for further analysis. DEGs between *S. aureus* infected and control BMDMs were identified using limma package (3.54.0). GO analysis of DEGs ($p$ value $< 0.05$, and $|Log_2(\text{fold change})| > 1$) was performed with the clusterProfiler package (Version 4.7.1) in R platform (Version 4.2.3, https://www.r-project.org/). Terms with adjusted $p$ values $< 0.05$ were considered significantly enriched. Heatmap was plotted using the pheatmap R package (version 1.0.12).

## Statistical analysis

Statistical analyses were performed using GraphPad Prism 8 (GraphPad Software, California, USA) or SPSS 26.0 (IBM, New York, USA). Quantifications were based on at least three independent experimental groups. Data were expressed as mean ± standard error of mean (SEM). Significant differences between two groups were evaluated by the Student's $t$ test or Mann-Whitney $U$ test. One-way ANOVA with Tukey's or Dunnett *post-hoc* test was used to compare multiple groups. A value of $p < 0.05$ was considered statistically significant.

## Supporting information

**S1 Fig. Expression levels of phosphorylated MEK1/2 (p-MEK1/2) in left femurs of *S. aureus* osteomyelitis mice and control mice.** (A) Representative images of immunofluorescence staining for F4/80$^+$ (red) and p-MEK1/2$^+$ (green) cells in femoral bone marrow. (B) Quantification of the number of F4/80$^+$p-MEK1/2$^+$ cells in the field of view. Scale bars, 50 μm. n = 5/group, Student's $t$ test.
(TIF)

**S2 Fig. PD0325901 treatment enhances the bactericidal function of BMDMs against *S. aureus*.** (A) Representative images of *S. aureus* colonies for phagocytosis assay. (B) Representative images of *S. aureus* colonies for the bactericidal assay. (C) Representative western blot images and (D) Quantification of relative levels of p22phox, p-p47phox, p47phox, p67phox, and p40phox. n = 5/group, one-way ANOVA with Tukey's *post-hoc* test was used.
(TIF)

**S3 Fig. Inhibition of TLR2 signaling using C29 blocks the upregulation of p-MEK1/2 levels by *S. aureus* infection.** (A) Representative images and (B) Western blot quantification for p-MEK1/2 and MEK1/2. n = 5/group, $^*p < 0.05$, $^{**}p < 0.01$, one-way ANOVA with Dunnett's *post-hoc* test was used.
(TIF)

**S4 Fig. Inhibition of EGFR signaling restores mitochondrial ROS (mtROS) production and increases bactericidal function.** (A) Representative images of western blots and (B) Quantification of p-EGFR levels relative to total EGFR in BMDMs infected by *S. aureus* (MOI = 10) for the indicated time. n = 7/group. $^*$ $p < 0.05$ versus control, one-way ANOVA with Tukey's *post-hoc* test used. BMDMs were pretreated with 10 μM erlotinib or vehicle (DMSO) for 1h, followed by *S. aureus* challenge for the indicated time. (C) Representative image of *S. aureus* colonies for phagocytosis assay. (D) Representative images of intracellular colonies of *S. aureus* for the bactericidal assay. BMDMs were infected with *S. aureus* at MOI of 10 for 1 h. After removing non-phagocytosed extracellular bacteria, cells were treated with 10 μM erlotinib or vehicle (DMSO) for the indicated time points. (E) Representative images and (F) Quantification of mtROS in BMDMs detected using MitoTracker Green and MitoSOX Red. Nuclei were stained with Hoechst 33342. Cells were infected with *S. aureus* (MOI = 10) for 1 h. After removing non-phagocytosed extracellular bacteria, cells were treated with 10 μM

erlotinib or vehicle (DMSO) for 12 h and 24 h. Scale bars, 10 μm. n = 3/group. ** $p < 0.01$, Student's *t* test was used. (G) Representative images of intracellular colonies of *S. aureus* for the bactericidal assay.
(TIF)

**S5 Fig. The effect of blocking MEK1/2 and EGFR signaling on the mRNA expression of IL-1β (A), IL-6 (B), and TNF-α (C).** BMDMs were infected with *S. aureus* at MOI of 10 for 1 h. After removing non-phagocytosed extracellular bacteria, cells were treated with 10 μM erlotinib, 1 μM PD0325901, or vehicle for 12 h. n = 8/group, * $p < 0.05$, ** $p < 0.01$, one-way ANOVA with Dunnett's *post-hoc* test used.
(TIF)

**S6 Fig. Blocking TLR2 does not affect phagocytosis but suppresses the bactericidal function of BMDMs.** (A) Representative image of intracellular colonies of *S. aureus* for phagocytosis assay. (B) Representative images of intracellular colonies of *S. aureus* for bacterial killing assay.
(TIF)

**S7 Fig. Blocking EGFR signaling improves bone structure in *S. aureus*-infected femurs.** (A) Representative images of immunofluorescence staining for p-EGFR in F4/80+ macrophages in the right femoral bone marrow of *S. aureus*-infected mice treated with erlotinib or vehicle, and control mice without infection. (B) Representative images of H&E staining for *S. aureus*-infected femurs from mice treated with erlotinib and vehicle. (C) Quantification of histopathological changes using the scoring system established by Smeltzer et al [51]. Blue arrows show trabecular bone with empty lacunae, green arrows show erosion in cortical bone, and black arrows show reactive new bone formation around cortical bone. Scale bar, 100 μm. n = 5/group, ** $p < 0.05$, Mann Whitney test.
(TIF)

**S1 Data. Source data used to generate all the graphs in this article.** File containing source data for all graphs generated in this article. Each sheet is labeled with the corresponding figure panel.
(XLSX)

## Acknowledgments

We thank Professor Liang Ping for English proofreading of the manuscript.

## Author Contributions

**Conceptualization:** Xianrong Zhang.

**Data curation:** Mingchao Jin, Xiaohu Wu, Jin Hu, Xianrong Zhang.

**Formal analysis:** Mingchao Jin, Xiaohu Wu, Jin Hu, Mankai Yang.

**Funding acquisition:** Xianrong Zhang.

**Investigation:** Mingchao Jin, Xiaohu Wu, Jin Hu, Yijie Chen, Bingsheng Yang, Chubin Cheng.

**Methodology:** Mingchao Jin, Xiaohu Wu, Jin Hu, Xianrong Zhang.

**Project administration:** Xianrong Zhang.

**Resources:** Xianrong Zhang.

**Supervision:** Xianrong Zhang.

**Validation:** Mingchao Jin, Xiaohu Wu, Jin Hu, Yijie Chen, Bingsheng Yang, Chubin Cheng.

**Visualization:** Mingchao Jin, Xiaohu Wu, Jin Hu, Mankai Yang.

**Writing – original draft:** Mingchao Jin, Xiaohu Wu, Xianrong Zhang.

**Writing – review & editing:** Xianrong Zhang.

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
