## [Decision Letter · Decision Letter 0]

3 Jan 2024

Dear Prof. Zhang,

Thank you very much for submitting your manuscript "EGFR-MEK1/2 cascade negatively regulates bactericidal function of bone marrow macrophages in mice with Staphylococcus aureus osteomyelitis" for consideration at PLOS Pathogens. As with all papers reviewed by the journal, your manuscript was reviewed by members of the editorial board and by several independent reviewers. In light of the reviews (below this email), we would like to invite the resubmission of a significantly-revised version that takes into account the reviewers' comments.

As noted by all three reviewers, this work seems well done and is a novel EGFR/MEK pathway examining host clearance of S. aureus, especially relevant than osteomyelitis model. However, there are several issues raised that must be addressed. Importantly, the pathway is based upon in vitro data using BMDMs and as shown in fig 3, insufficient numbers of replicates were performed - with substantial variability to support the authors' conclusions. Similarly, each reviewer has issues that must be addressed as clearly delineated below, including reliance upon a sole pharmacological MEK inhibitor. It is also necessary to clarify the presentation of the data, usage of English syntax is not accurate, abbreviations are difficult to find, and some of the conclusions are overstated.

We cannot make any decision about publication until we have seen the revised manuscript and your response to the reviewers' comments. Your revised manuscript is also likely to be sent to reviewers for further evaluation.

Sincerely,

Alice Prince

Academic Editor

PLOS Pathogens

Michael Otto

Section Editor

PLOS Pathogens

Kasturi Haldar

Editor-in-Chief

PLOS Pathogens

orcid.org/0000-0001-5065-158X

Michael Malim

Editor-in-Chief

PLOS Pathogens

orcid.org/0000-0002-7699-2064

As noted by all three reviewers, this work seems well done and is a novel EGFR/MEK pathway examining host clearance of S. aureus, especially relevant than osteomyelitis model. However, there are several issues raised that must be addressed. Importantly, the pathway is based upon in vitro data using BMDMs and as shown in fig 3, insufficient numbers of replicates were performed - with substantial variability to support the authors' conclusions. Similarly, each reviewer has issues that must be addressed as clearly delineated below, including reliance upon a sole pharmacological MEK inhibitor. It is also necessary to clarify the presentation of the data, usage of English syntax is not accurate, abbreviations are difficult to find, and some of the conclusions are overstated.

Reviewer's Responses to Questions

**Part I - Summary**

Reviewer #1: In the submitted manuscript Wu et al demonstrate MEK1/2 putatively inhibits Staphylococcal killing by macrophages, and that blocking of this pathway enhances outcomes in a osteomyelitis model of infection. The manuscript is marked by impressive outcomes in the preclinical model that is used. However, small cohorts used for in vitro studies are problematic and strongly diminish the rigor of the conclusions made by the authors. These should be adressed

Reviewer #2: The manuscript “EGFR-MEK1/2 cascade negatively regulates bactericidal function of bone marrow macrophages in mice with Staphylococcus aureus osteomyelitis” by Wu et al. investigates the role of the MEK1/2 pathway in a Staphylococcus aureus osteomyelitis in a mouse infection model, along with transcriptomic analysis and in vitro methodology. They demonstrate that S. aureus osteomyelitis infection in mice activates the EGFR-MEK1/2 cascade resulting in a reduction in mtROS levels, dampening the bactericidal abilities of macrophages. This has interesting potential if blocking such pathways can be combined with antibiotic treatment in vivo, but further research is required to establish this. In general this is a thoroughly researched manuscript but in my opinion requires revisions before accepting for publication, including expanding upon how some conclusions have been made. A lot of data is shown but I suggest some of it can be moved to supplementary figures to make the main figures clearer to interpret. My comments to the authors are detailed below.

Reviewer #3: The manuscript written by Wu, X. et al. submitted to Plos Pathogens studies the role of the EGFR/MEK pathway in S. aureus killing by macrophages and conclude that the bacterium upregulates this signaling pathway to increase its survival within macrophages that likely contributes to its persistent infection seen in osteomyelitis. The authors established an in vivo model of S. aureus osteomyelitis in mice and showcase the beneficial in vivo effects of the MEK inhibitor, PD0325901. The report also investigates the role of EGFR, mitoROS and Chek2, a positive regulator for mtROS production. The data are interesting and represent a novel mechanism by which S. aureus seems to promote its persistence in host macrophages. However, the detailed signaling mechanism is largely based on in vitro experiments using BMDMs infected with S. aureus. Additional in vivo experiments focusing on key points of the signaling mechanisms revealed in vitro would further increase the impact of the work. Also, exploring phagosomal, not mitochondrial, ROS production is missing and is needed to understand the whole picture. Please find below my major and minor comments.

**Part II – Major Issues: Key Experiments Required for Acceptance**

Reviewer #1: The greatest issue I have for the paper is the paucity of replicates of the in vitro work. The authors have multiple figures, some which are critical to support the conclusions of the work, that have n=3-4 (e.g. Fig 3A, 3F, 4I, 4K…). These are in vitro experiments and the n= should be at least 8.

In addition killing rates in Fig 4 and Fig E, should be demonstrated in fashion similar to Fig 3F, and multiple time points are shown

Why is Erlotinib and 325901 not tested together? This is relevant.

Reviewer #2: Figure 2: In lines 144-146 you state “Our data clearly showed that the levels of p-MEK1/2 were upregulated predominantly in F4/80+ macrophages surrounding the abscess”. Did you investigate p-MEK1/2 levels in F4/80+ macrophages away from the abscess, for example in the left femur of the S. aureus infected mice, to support your statement that MEK1/2 levels are increased specifically at the abscess site? Please include this data if so, otherwise adjust your conclusions accordingly to e.g. “Our data clearly showed that the levels of p-MEK1/2 were upregulated in F4/80+ macrophages of S. aureus infected mice compared to uninfected controls”.

Reviewer #3: - The authors should show that erlotinib reduces EGFR phosphorylation as they expect it

- While the investigators indicate the involvement of the EGFR/MEK/mROS and Chek2 signaling in S. aureus persistence in macrophages in vitro, most of these data were not proven in vivo, in their osteomyelitis model. Testing some of these inhibitors in their mouse model would be important to increase the clinical relevance of their findings. Alternatively, the authors could test their model in mice deficient in Tlr2 or Egfr.

- Mitochondrial ROS production has been implicated as important in microbial killing of phagocytes including macrophages, its exact mechanism of action remains largely unclear. The investigators should also comment and provide data on a much better accepted and better understood killing mechanism of S. aureus, phagocytic ROS production. It is known to be critical for S. aureus killing. Please look into phagosomal ROS production or phosphorylation of NADPH oxidase components in S. aureus -phagocytosing macrophages.

**Part III – Minor Issues: Editorial and Data Presentation Modifications**

Reviewer #1: Line 79. Authors reference for S. aureus survival within macrophages as mechanism for chronic / relapsing infections. One reference, the review, is primarily focused on neutrophils. The other reference on the impact of PD-L1 signaling – neither paper specifically suggests that intracellular survival within macrophages is the mechanism for chronic infection. This statement should be modified, or references updated.

Line 113+. Lots of acronyms that are not described before. Their meaning can be found in the figure legend but should be in the text. Accordingly, Fig 1 is very complicated, and it is not clear what the readers should be looking at so as to gleam meaningful positive outcomes. This could be better described.

Line 176. Why is TLR2 data not included as part of the findings presented here with the other TLRs? When reading this, I immediately asked myself what about TLR2 – why is it left out? Only to find it was examined later. The role of TLR2 helps to explain why EGFR inhibition is partial (line 188).

Reviewer #2: Supplementary figure 1: Please elaborate upon what the different treatments (5 μM ML130, 5 μM GSK717, 0.5 μM GSK2983, 10 μM NOD-IN-1, 5 μM CU-CPT9a, 1 μM E6446) target, explaining why they have been selected for this assay. Expand upon the statement in lines 176-177 “Results showed that neither TLR7/8/9…” to explain what the assay demonstrates. Please detail what the control is – is this vehicle treated too?

Please detail what the vehicle is and whether or not the same vehicle is used throughout all the assays in this manuscript.

Line 95: Please change “improved bone destruction” to “rescued bone destruction”.

Figure 1 (and all subsequent figures): Please state what error bars indicate e.g. standard deviation, standard error of mean, and consistently show error bars in both directions.

Figure 1K (and throughout the rest of the figures): Please make the scale bars thicker and therefore easier to see.

Figure 3G, 4L, 5F: Please state what the stain shown in blue is.

Supplementary figure 2B: Please show the control group on the graph for example by adjusting the Y axis to make it visible.

Line 226: “Fig 6A and 5B” should read 6A and 6B.

Figure 6E: I do not think splitting the images into Chek2/DAPI and Chek2/MitoTracker Green adds anything to this figure, particularly as the DAPI staining is also shown in the Chek2/MitoTracker Green panel. I suggest removing the top panel and relabelling the remaining panel as Chek2/MitoTracker Green/DAPI.

Line 254: Correct the spelling of “Salmaonella” to “Salmonella”.

Figures 3/4/5: I suggest the images of agar plates for the phagocytosis/bacterial killing assays could be made into supplementary figures instead, particularly as quantified data is shown in these figures too. This could allow for other panels of the figures to be made larger and overall the figures less crowded.

Line 456: It states that Hoechst33324 was used to label nuclear DNA, but otherwise in the manuscript DAPI is specified. Please clarify which is the case and correct Hoechst33324 to 33342 if it was used.

Reviewer #3: Minor comments

- The manuscript should be edited by a native English speaker

- Line 117-118. The T.b.. T.h. and P.f. abbreviations should be introduced at their first mentioning.

- Line 112. If I am reading it correctly, the authors refer to the PD compound as an adjuvant to gentamicin. Vaccines are not used in this paper, so it remains unclear to me why the PD compound is referred to in the paper as an adjuvant?

- The authors should explain why was the use of gentamicin needed in the osteomyelitis model? Gentamicin was used to prevent a systemic S. aureus infection or infections due to potential contaminating microorganisms? Please add clarification about this.

- Data shown in Figures 3C and D were only done three times and show a wide spread. Therefore, more experiments should be performed to support the authors’ related claim that the PD inhibitor does not affect S. aureus phagocytosis by BMDMs.

-

PLOS authors have the option to publish the peer review history of their article (what does this mean?). If published, this will include your full peer review and any attached files.

Reviewer #1: No

Reviewer #2: **Yes: **Amy K Tooke

Reviewer #3: **Yes: **Balázs Rada

Figure Files:

Data Requirements:

Please note that, as a condition of publication, PLOS' data policy requires that you make available all data used to draw the conclusions 

---

## [Editor Report · Decision Letter 1]

4 Jul 2024

Dear Prof. Zhang,

Thank you very much for submitting your manuscript "EGFR-MEK1/2 cascade negatively regulates bactericidal function of bone marrow macrophages in mice with Staphylococcus aureus osteomyelitis" for consideration at PLOS Pathogens. As with all papers reviewed by the journal, your manuscript was reviewed by members of the editorial board and by several independent reviewers. The reviewers appreciated the attention to an important topic. Based on the reviews, we are likely to accept this manuscript for publication, providing that you modify the manuscript according to the review recommendations.

The authors have appropriately responded to the reviewers criticisms with new data, increased numbers of replicates and clarification of the involvement of EGFR and TLR2 pathways. The grammar and syntax is also much improved however with remaining errors in the following lines that should be corrected:

l. 82, l161, l165, 1. 269 (should read "mice with implant associated osteomyelitis) - as well as many others remaining - usually in the lack of articles such as "a" or "the". Otherwise the manscript is a nice contribution that very thoroughly details the role of this MEK/EGFR pathway in macrophage killing of S. aureus in a clinically relevant model.

Sincerely,

Alice Prince

Academic Editor

PLOS Pathogens

Michael Otto

Section Editor

PLOS Pathogens

Michael Malim

Editor-in-Chief

PLOS Pathogens

orcid.org/0000-0002-7699-2064

The authors have appropriately responded to the reviewers criticisms with new data, increased numbers of replicates and clarification of the involvement of EGFR and TLR2 pathways. The grammar and syntax is also much improved however with remaining errors in the following lines that should be corrected:

l. 82, l161, l165, 1. 269 (should read "mice with implant associated osteomyelitis) - as well as many others remaining - usually in the lack of articles such as "a" or "the". Otherwise the manscript is a nice contribution that very thoroughly details the role of this MEK/EGFR pathway in macrophage killing of S. aureus in a clinically relevant model.

Reviewer Comments (if any, and for reference):

Figure Files:

Data Requirements:

Reproducibility:

References:

---

## [Editor Report · Decision Letter 2]

22 Jul 2024

Dear Prof. Zhang,

We are pleased to inform you that your manuscript 'EGFR-MEK1/2 cascade negatively regulates bactericidal function of bone marrow macrophages in mice with Staphylococcus aureus osteomyelitis' has been provisionally accepted for publication in PLOS Pathogens.

Best regards,

Michael Otto

Section Editor

PLOS Pathogens

Michael Otto

Section Editor

PLOS Pathogens

Michael Malim

Editor-in-Chief

PLOS Pathogens

orcid.org/0000-0002-7699-2064
---

## [Editor Report · Acceptance letter]

28 Jul 2024

Dear Prof. Zhang,

We are delighted to inform you that your manuscript, "EGFR-MEK1/2 cascade negatively regulates bactericidal function of bone marrow macrophages in mice with Staphylococcus aureus osteomyelitis," has been formally accepted for publication in PLOS Pathogens.

Best regards,

Michael Malim

Editor-in-Chief

PLOS Pathogens

orcid.org/0000-0002-7699-2064